# Dietary Behaviors, Serum 25(OH)D Levels and Quality of Life in Women with Osteoporotic Disorders

**DOI:** 10.3390/ijerph192417023

**Published:** 2022-12-18

**Authors:** Małgorzata Godala, Ewa Sewerynek, Ewelina Gaszyńska

**Affiliations:** 1Department of Nutrition and Epidemiology, Medical University of Lodz, Zeligowskiego Street 7/9, 90-752 Lodz, Poland; 2Department of Endocrine Disorders and Bone Metabolism, Medical University of Lodz, 90-752 Lodz, Poland

**Keywords:** quality of life, Qualeffo-41 Questionnaire, osteoporotic disorders, 25(OH)D, nutrition knowledge

## Abstract

Data obtained in recent years clearly demonstrate the aging process of European populations. Consequently, the incidence of osteoporosis has been rising. The aim of this study is to assess the quality of life (QoL) of women with osteoporosis. A total of 260 women participated in this study. The patient group consisted of 170 women with osteoporotic disorders. The control group consisted of 90 healthy women. Participants’ quality of life was measured with the Qualeffo-41 Questionnaire. The total 25(OH)D concentration level was assessed with an assay using the chemiluminescent immunoassay. To assess the pain level, the Visual Analogue Scale (VAS) was used. To assess dietary behaviors, data were obtained by a 13-item Food Frequency Questionnaire. To assess the nutrition knowledge of participants, the Beliefs and Eating Habits Questionnaire was used. Based on the frequency of food intake, participants were classified into three patterns of behavior, i.e., Prudent, Western, and Not Prudent-Not Western. The patients assessed their quality of life as average (36.6 ± 19.9 points). The most favorable scores were obtained in the domains of “Ability to do jobs around the house” and “Mobility”. The worst rated domain among the respondents was “Mental function”. There were significant differences identified in quality of life depending on diet, nutritional knowledge, comorbidities and occurrence of fractures in the subjects. The individuals in the “Prudent” group reported a significantly higher quality of life as compared to the “Not Prudent-Not Western” and “Western” groups and those with high nutritional knowledge as compared to those with moderate and low. Lower quality of life was also observed among women with comorbidities and with bone fractures. Depending on serum 25(OH)D levels, poorer quality of life was characterized women with vitamin D deficiency. Patient education, implementation of effective methods aimed at alleviating pain and maintaining the optimal concentration of vitamin D can help improve the quality of life in patients with osteoporotic disorders.

## 1. Introduction

Data obtained in recent years clearly demonstrate the ageing process of the Polish and European populations, which results from increased life expectancy, among others. Consequently, the incidence of osteoporosis has also been rising in Poland [1]. The most common complications of osteoporosis are bone fractures, which are associated with pain and cause limitations of both physical and social functions regarded as the determinants of health-related quality of life (HRQoL) [2,3,4]. Due to demographic changes in the Polish and European populations, the number of bone fractures is expected to increase by 25% in the coming years [2,5]. For many patients, it means loss of independence, more severe pain and deterioration in health-related quality of life, which can affect the patient’s mental condition and impose limitations in social life [6,7].

A risk factor for osteopenia and osteoporosis is vitamin D deficiency. This deficiency, which leads to increased parathyroid hormone (PTH) secretion and decreased absorption of calcium from the digestive tract with subsequent acceleration of bone turnover, development of osteomalacia and myopathy, has been found in more than 50% of adolescents and adults and 70–80% of the elderly in Europe [1]. In the adult population, vitamin D deficiency is one of the main factors leading to a higher risk of fractures due to a lower mechanical strength of the skeleton and increased number of falls. This is the reason for investigating vitamin D intake, including supplements and diet sources in these patients. The consequent fractures are responsible for higher morbidity and mortality among the elderly. Recent studies suggest that the beneficial effect of vitamin D in the prevention of falls and fractures is observed with a daily intake of 700–800 IU of cholecalciferol, and that the adequate supply of vitamin D, as measured by serum 25(OH)D concentration, should be 30 ng/mL [8,9].

Studies have shown that bone fractures result in the deterioration of quality of life [3,4,10]. Additionally, this type of injury is associated with the development of other complications, such as disability, depression, cardiovascular diseases, stroke and sleep disorders. Consequently, this leads to increased mortality [10,11,12]. The growing number of bone fractures in the course of osteoporosis, especially in developed countries, is a serious health and economic problem [10,11,12,13]. It is estimated that, in Europe, the cost of osteoporosis treatment will increase by 27% in 2025 as compared to 2010 [10,14]. Studies have highlighted that osteoporosis, in comparison to other non-communicable diseases, is rarely adequately subsidized by state authorities and healthcare providers. This trend is also observed in Poland, where there are only 4.3 Dual-energy X-ray Absorptiometry (DXA) densitometers per million population [10]. For comparison, the most favorable situation is in Belgium, where there are as many as 53 densitometry apparatuses per million inhabitants. This situation causes difficulties in access to diagnosis, adequate monitoring and treatment of the disease [10].

The aim of this study is to assess the quality of life (QoL) of women with osteoporosis in relation to their dietary knowledge and behaviors, presence of pain and serum 25(OH)D levels.

## 2. Materials and Methods

### 2.1. Study Group

The study group comprised of 260 women, including 170 women with osteoporotic disorders (aged 48–74 years, the mean age 61.7 ± 11.8 years), recruited from the Department of Endocrine Disorders and Bone Metabolism, Chair of Endocrinology, Medical University of Lodz. Bone density measurements were obtained and ISCD (International Society for Clinical Densitometry) criteria was used to define osteoporosis. The control group consisted of 90 clinically healthy women (aged 46–72, the mean age 59.4 ±11.8 years), without the above disorder. Ninety women without osteoporosis).

The study was approved by the Bioethics Committee of the Medical University of Lodz.

### 2.2. Anthropometry

All subjects had their BMI (the body mass index) and waist to hip ratio (WHR) determined. The body composition was assessed with electrical bioimpedance analysis (BIA) using the Bodystat 1500 MDD apparatus.

### 2.3. Quality of Life Assessment

Qualeffelo-41 Questionnaire was used to determine patients’ quality of life. Seven components, including pain, the activities of daily living, the ability to do jobs around house, mobility, leisure and social activities, general health perception, and mental function were measured, with possible scores between 0–100 points. A lower mean score indicates a higher participants’ quality of life [6,7,15,16].

### 2.4. Pain Assessment

Visual Analogue Scale (VAS) was used to assess pain level. The 100-mm line, with “0” meaning “no pain” and “100” meaning “the greatest pain” was used. The following cut points have been assigned: No pain (0–4 mm), mild pain (5–44 mm), moderate pain (45–74 mm), and severe pain (75–100 mm) [17].

### 2.5. Nutrition Assessment

To assess dietary behaviors, data were obtained by a 13-item Food Frequency Questionnaire (FFQ) using the previous week/month as a reference period. Participants were asked to assess frequency of consumption as servings per day or per week for each food group.

Based on the frequency of food intake, participants were classified into three patterns of behavior, i.e., Prudent, Western, and Not Prudent-Not Western. The subjects in the Prudent group were characterized by a high intake of fruits, vegetables, whole grains, dairy products and fish. The Western group was characterized by a high intake of fast-food products, white bread, red meat, sweets, and sweetened beverages. The Not Prudent-Not Western group was characterized by a low intake of vegetables, fruits, fish, fast foods, sweetened beverages, and sweets.

The Beliefs and Eating Habits Questionnaire (part C) created by the Polish Academy of Science was used to determined patients’ nutrition knowledge [18]. Each correct answer was scored with 1 point, and incorrect or “I don’t know” answers were scored with 0 points. Three levels of nutrition knowledge categories were classified: Insufficient (0–8 points), sufficient (9–16 points) and high (17–25 points).

### 2.6. Vitamin D Assessment

Fasting blood samples were used to determine the total 25(OH)D concentration level with an assay using the chemiluminescent immunoassay (CLIA) methodology. A serum 25(OH)D concentration of at least 30 ng/mL was considered normal, whereas the level below 30 ng/mL was considered insufficient (deficient concentration) [9].

### 2.7. Statistical Analysis

A statistical analysis was performed using the Statistica v.13 programme. Descriptive statistics with determination of the mean and standard deviation or median and interquartile range were made. The normal distribution was determined using the Shapiro–Wilk test. When the analyzed variables appeared to be incompatible with the normal distribution, the Mann–Whitney test was used to compare the groups. The Spearman’s rank order correlation with determination of the Spearman’s R coefficient was used to determine the correlation between variables. The odds ratios (OR) with 95% confidence intervals (95% CI) were measured and crude models were created. Wald statistics were used to assess the significance of ORs. *p* < 0.05 was considered as significant for all tests used.

## 3. Results

There were no significant differences between the patients and the control group in terms of age, place of residence, smoking and presence of other diseases. Among the patients, 83 women lived in rural areas (48.8%), 21 smoked cigarettes (12.4%) and 65 (38.2%) had concomitant diseases requiring treatment, such as hypertension, endocrine disorders and rheumatoid arthritis. Bone fractures suffered in the past were declared by 71 study subjects (41.8%), significantly more often than in the control group. Fractures of the upper limbs, lower limbs and the hip bone were among the most frequently reported ones. The majority of patients (51.2%) had normal body weight as assessed by BMI. On the visual analog scale (VAS), on average the patients rated their pain as mild (4.42 ± 2.73 points). The score was significantly higher than in the control group (Table 1).

Using the QUALEFFO-41 scale, the patients assessed their quality of life as average (36.6 ± 19.9 points). The score was significantly higher as compared to the control group. Of the seven domains analyzed, the most favorable scores were obtained in the domains of “Ability to do jobs around the house” (27.7 ± 17.2 points) and “Mobility” (28.4 ± 15.1 points). The worst rated domain (48.4 ± 18.6 points) among the respondents was “Mental function” (Table 2).

When the quality of life of the study participants with osteoporosis was compared in relation to their place of residence and smoking, no significant differences were found. Whereas there were significant differences identified in quality of life depending on diet, nutritional knowledge, comorbidities and occurrence of fractures in the subjects. Thus, the individuals in the “Prudent” group reported a significantly higher quality of life as compared to the “Not Prudent-Not Western” and “Western” groups (22.7 ± 15.2 vs. 32.8 ± 12.7 vs. 52.3 ± 11.5, *p* = 0.000, respectively) and those with high nutritional knowledge as compared to those with moderate and low knowledge (27.1 ± 11.9 vs. 35.2 ± 14.8 vs. 48.7 ± 16.5, *p* = 0.000, respectively). Significant differences were also observed in relation to occurrence of other diseases requiring treatment and bone fractures. Lower quality of life was also observed among women with comorbidities as compared to the women without other diseases (40.8 ± 19.2 vs. 35.6 ± 12.9, *p* = 0.000), as well as in the group of subjects with bone fractures as compared to those who did not suffer such injuries (51.7 ± 15.3, 25.1 ± 17.9, *p* = 0.000) (Table 3).

Correlations between the domains analyzed in the questionnaire were also assessed. The coefficients ranged from 0.305 to 0.675. The strongest correlation in the patients was found between “Activities of daily living” and “Leisure and social activities” (r = 0.675). High correlation coefficients were also observed between “Mobility” and “Pain”, “Activities of daily living” and “Pain”, “Ability to do jobs around the house” and “Leisure and social activities”, and “General health perception” and “Mental function” (Table 4).

There were statistically significant changes observed in the quality of life of the subjects depending on serum 25(OH)D levels, with higher values, and thus poorer quality of life, in the group of women with vitamin D deficiency. Whereas, as for the domains “Activities of daily living”, “Ability to do jobs around the house” and “Pain”, the differences were statistically insignificant (Table 5).

Correlations were found between quality of life and age of the patients (r = 0.52, *p* < 0.05), BMI (r = −0.21, *p* < 0.05), and pain assessed using the VAS (r = 0.47, *p* < 0.05). The relations were confirmed for each domain of the QUALEFFO-41 scale. Additionally, an association was found between the patients’ quality of life and serum 25(OH)D levels (r = −0.24, *p* < 0.05). The relationships were statistically significant for the following domains: “Mental function”, “General health perception”, “Leisure and social activities” and “Mobility” (Table 6).

## 4. Discussion

In our study, the subjects with osteoporosis assessed their quality of life as moderate, significantly worse than the women in the control group. However, the differences were shown between the various domains assessed using the QUALEFFO-41 scale. The lowest quality of life was shown in the “Mental function”. In a study by Gorczewska et al., conducted on a group of 198 postmenopausal women with osteoporosis, a moderate level of quality of life was found, as measured by the QUALEFFO-41 scale. Similar to the results of our study, the domain that was assessed most unfavorably was “Mental function” [19]. In a study by Baczyk et al., patients with osteoporosis assessed their quality of life using the QUALEFFO-41 scale at 28.9 ± 118 points, and women with osteopenia at 26.7 ± 12.2 points. In contrast, in this study, unlike in our work, the lowest-rated domain was “General health perception” [20]. Similar data were obtained in a study by Nawrat-Szołtysik et al., which included a group of men with osteoporosis. The respondents described their quality of life as moderate (Me = 38 points), and the lowest-rated domain was “Leisure and social activities” [21]. Similarly, in a study by Drozd et al., postmenopausal women with osteoporosis scored an average of 49.38 points on quality of life, while the worst-rated domain was “General health perception” [22]. In our study, there were correlations between the specific domains assessed using the QUALEFFO-41 questionnaire, with the strongest one found for “Activities of daily living” and “Leisure and social activities”. Comparable correlations were also reported in other studies [22,23,24].

In the present study, we observed significantly lower levels of quality of life depending on the dietary pattern, nutritional knowledge, co-morbidities and occurrence of fractures in the subjects. Quality of life was assessed to be higher by women following a rational dietary model, those with sufficient and high nutritional knowledge, and patients without comorbidities or a history of fractures. To the best of our knowledge, this is the first study to analyze the relationship between dietary behaviors and quality of life in osteoporotic patients. However, there are data indicating that the diagnosis prompts osteoporotic patients to change their lifestyle to a health-promoting one [25]. It is probably associated with improved dietary behavior and better nutritional knowledge, and thus has a positive impact on quality of life. In contrast, papers evaluating the impact of fracture incidence on the quality of life of osteoporotic patients are numerous [26,27,28,29,30,31,32,33]. In our study, a correlation was found between occurrence of fractures in the past and poorer quality of life in the patients. Similar data were obtained in a study by Lips et al. [16] and Gold et al. [26], in which presence of fractures was a factor that significantly worsened the quality of life of the subjects. Comparable data were obtained in a study by Baczyk et al. [20]. Also, in a study by Kuru et al., a history of fractures resulted in a worse quality of life of patients with osteoporosis [27]. In contrast, in a study by Drozd et al., subjects with a history of bone fractures assessed their quality of life lower as compared to women without fractures, however, the differences were not statistically significant [22]. In a study by Gorczewska et al., the respondents with a history of any bone fractures obtained worse scores of their quality of life on the QUALEFFO-41 scale (42.05 ± 17.31 points) than those without fractures (36.25 ± 15.40 points). The study further found that quality of life was rated lower by women with femoral neck fractures than those with fractures of other bones [19]. Similar data were obtained in a study by Adachi et al., in which, apart from the deterioration in quality of life associated with the occurrence of fractures, the type of fracture was shown to be a determinant of a patient’s quality of life. Thus, quality of life was most severely impaired by femoral neck and pelvic fractures in women and femoral neck fractures in men [33]. In our study, a similar analysis was not possible due to the insufficient number and diversity of patients with fractures.

The study found that the determinants of quality of life in the subjects were age and BMI. Similar data were obtained in other studies. A study by Lips et al., demonstrates that quality of life deteriorated with age [16]. The Adelphi US Osteoporosis Disease Specific Programme showed that the determinants of quality of life in patients with osteoporosis were the subjects’ age and BMI [31]. However, a study by Drozd et al., found that the quality of life of postmenopausal women with osteoporosis deteriorated with age, while this relationship was not confirmed for BMI [22].

Additionally, we showed a negative effect of the level of perceived pain on the quality of life in osteoporotic patients. These data were confirmed in a study by Gorczewska et al., which found a positive correlation with the level of perceived pain in all domains of the QUALEFFO-41 scale [19]. Additionally, in a study by Stanghelle et al., conducted on a group of women with osteoporosis, the level of perceived pain was a factor significantly affecting the subjects’ quality of life in all domains of the QUALEFFO-41 scale [34]. In contrast, in a study by Nawrat-Szołtysik, pain was not a factor significantly affecting quality of life of men with osteoporosis as measured by the QUALEFFO-41 scale, both in total and in individual domains, except for “General health perception” [21]. It has been also reported that osteoporotic fractures have the negative effect on respiratory function and impairment, which surely negatively affects the patients’ quality of life [35].

The results of our study indicate a relation between vitamin D nutrition and quality of life of osteoporotic women. The subjects with normal 25(OH)D concentrations rated their quality of life higher as compared to those with 25(OH)D deficiency. Additionally, we found an inverse correlation between serum 25(OH)D concentration and QUALEFFO-41 scale scores, thus demonstrating that the higher the serum 25(OH)D concentration, the better the subjects’ quality of life. In the literature, there are only few papers analyzing the relationship between the quality of life in patients with osteoporosis and vitamin D concentrations. In a study by Korkmaz et al., women with postmenopausal osteoporosis and serum 25(OH)D concentrations below 15 ng/dL had a significantly worse quality of life scores as compared to women with 25(OH)D concentrations ≥ 15 ng/dL. Similarly as in our study, an inverse relationship between serum vitamin D concentration and some domains of the QUALEFFO-41 scale has been shown. In this study, 25(OH)D concentration inversely correlated with quality of life in the domains of “Mobility” and “General health perception” [23]. A study by Ohta et al., conducted in a group of 1585 patients with osteoporosis showed that serum 25(OH)D levels were an independent factor affecting the subjects’ quality of life. They found that higher serum 25(OH)D levels correlated with a better quality of life, and that patients with normal vitamin D status rated their QoL significantly higher as compared to patients with 25(OH)D deficiency [36]. Due to its pleiotropic effects, vitamin D has been discussed in the literature as a factor that regulates cell proliferation, functioning of the immune system, blood circulation and physical activity. There are works which demonstrate a relation between vitamin D deficiency and falls, especially among the elderly. Studies have shown an association between low serum 25(OH)D levels and a higher incidence of fractures [37,38]. Due to the fact that fractures are the most common complication of osteoporotic lesions, it can be assumed that vitamin D levels indirectly affect the perceived quality of life in these patients, as shown in our survey. It is worth mentioning that vitamin D was recently reported as a risk factor not only for osteoporosis, but also for body mass alterations, including sarcopenia and obesity in the novel described osteosarcopenic obesity phenotype [39].

The conducted study has some limitations. The study group was gender-homogenous, therefore the results cannot be generalized to the entire population of osteoporotic patients. Additionally, the incidence of falls and fracture risk, which have a significant impact on quality of life in these patients, were not verified.

## 5. Conclusions

This study found that health-promoting dietary behaviors, age, BMI, pain ailments and serum 25(OH)D levels were independent factors affecting the quality of life of the women with osteoporosis. Therefore, patient education, implementation of effective methods aimed at alleviating pain and maintaining optimal concentration of vitamin D can help improve the quality of life of patients with osteoporotic disorders.

## Figures and Tables

**Table 1 ijerph-19-17023-t001:** General characteristics of study population.

Characteristics	Patients *n* = 170	Controls *n* = 90	
	Mean ± SD/*n* (%)		*p*
Age [years]	61.7 ± 11.8	59.4 ±10.3	0.419
Rural	83 (48.8)	47 (52.2)	0.562
Current smokers	21 (12.4)	10 (11.1)	0.375
Fracture			
Yes	71 (41.8)	12 (13.3)	0.000
Coexisting diseases			
Yes	65 (38.2)	33 (36.7)	0.082
Waist circumference [cm]	85.7 ± 12.9	86.5± 11.5	0.127
Waist hip ratio (WHR)	0.8 ± 0.2	0.9 ± 0.1	0.093
BMI [kg/m^2^]	26.2 ± 7.5	27.1 ± 5.9	0.216
BMI categories			
Underweight	19 (11.2)	10 (11.1)	0.065
Normal weight	87 (51.2)	46 (51.1)	0.076
Overweight	38 (22.3)	21 (23.4)	0.421
Obesity	26 (15.3)	13 (14.4)	0.286
Body composition			
Body mass [kg]	65.6 ± 11.3	67.4 ± 9.7	0.432
Fat free mass [kg]	40.3 ± 8.7	41.1 ± 9.1	0.481
Fat mass [kg]	26.4 ± 10.8	27.1 ± 9.8	0.662
Fat mass [%]	39.7 ± 15.9	38.5 ± 12.4	0.713
Pain [points]	4.4 ± 2.7	3.1 ± 1.8	0.000
Pain categories			
No pain	16 (9.4)	35 (38.9)	0.000
Mild	112 (65.9)	40 (44.4)	0.000
Moderate	35 (20.6)	13 (14.4)	0.000
Severe	7 (4.1)	2 (2.2)	0.000
25(OH)D [ng/mL]	25.3 ± 10.9	27.9 ± 9.8	0.063
25(OH)D deficiency	114 (67.1)	59 (65.6)	0.091
Dietary patterns	*n* (%)		
Prudent	47 (27.6)	28 (31.1)	0.138
Not Prudent–Not Western	83 (48.8)	43 (47.8)	0.618
Western	40 (23.5)	19 (21.1)	0.225
Nutrition knowledge	*n* (%)		
Insufficient	62 (36.5)	31 (34.4)	0.043
Sufficient	76 (44.7)	45 (50)	0.017
High	32 (18.8)	14 (15.6)	0.021

**Table 2 ijerph-19-17023-t002:** Quality of life of study patients measured with Qualeffo-41.

QUALEFFO-41 Domains	Patients	Controls	*p*
	Mean ± SD		
Activities of daily living	30.5 ± 20.8	26.9 ± 17.2	0.000
Ability to do jobs around the house	27.7 ± 17.2	24.2 ± 13.8	0.000
Mobility	28.4 ±15.1	26.1 ± 16.2	0.000
Pain	30.7 ± 22.7	22.9 ± 17.1	0.000
General health perception	43.6 ± 16.3	40.3 ± 14.1	0.000
Leisure and social activities	40.9 ± 20.4	38.1 ± 10.5	0.000
Mental function	48.4 ± 18.6	39.7 ± 16.5	0.000
Total	36.6 ± 17.9	30.8 ± 12.5	0.000

**Table 3 ijerph-19-17023-t003:** The average indexes of quality of life and dietary patterns, nutrition knowledge, fractures, and coexisting diseases in women with osteoporosis.

QUALEFFO-41 Domains	Total [Points] (Mean ± SD)	*p*
Dietary patterns of behavior		
Prudent	22.7 ± 15.2	0.000
Not Prudent–Not Western	32.8 ± 12.7	
Western	52.3 ± 11.5	
Nutrition knowledge		
Insufficient	48.7 ± 16.5	0.000
Sufficient	35.2 ± 14.8	
High	27.1 ± 11.9	
Fractures		
Yes	51.7 ± 15.3	0.000
No	25.1 ± 17.9	
Coexisting diseases		
Yes	40.8 ± 19.2	0.000
No	35.6 ± 12.9	

**Table 4 ijerph-19-17023-t004:** The correlation between domains of the Qualeffo-41 in women with osteoporosis (Spearman’s correlation coefficient, *p*).

	Activities of Daily Living	Ability to Do Jobs around the House	Mobility	Pain	General Health Perception	Leisure and Social Activities	Mental Function
Activities of daily living	1	0.565, 0.001	0.497, 0.001	0.612, 0.000	0.571, 0.000	0.675, 0.000	0.552, 0.001
Ability to do jobs around the house	-	1	0.552, 0.001	0.584, 0.000	0.558, 0.000	0.625, 0.000	0.357, 0.001
Mobility	-	-	1	0.631, 0.000	0.305, 0.001	0.472, 0.001	0.378, 0.001
Pain	-	-		1	0.402, 0.000	0.481, 0.000	0.351, 0.005
General health perception	-	-	-	-	1	0.398, 0.001	0.615, 0.000
Leisure and social activities	-	-	-	-	-	1	0.598, 0.001
Mental function	-	-	-	-	-	-	1

**Table 5 ijerph-19-17023-t005:** Quality of life in patients according to vitamin D level.

QUALEFFO-41 Domains	25(OH)D < 30 ng/mL *n* = 114	25(OH)D ≥ 30 ng/mL *n* = 56	*p*
Activities of daily living	33.2 ± 15.6	30.5 ± 19.2	0.126
Ability to do jobs around the house	28.6 ± 17.1	25.8 ± 15.9	0.118
Mobility	30.1 ± 19.3	26.0 ± 16.9	0.031
Pain	40.3 ± 20.3	29.5 ± 18.3	0.123
General health perception	48.1 ± 14.9	38.7 ± 20.3	0.041
Leisure and social activities	38.2 ± 15.7	50.5 ± 13.9	0.033
Mental function	45.6 ± 17.2	51.3 ± 19.5	0.025
Total	41.5 ± 15.9	32.8 ± 16.7	0.022

**Table 6 ijerph-19-17023-t006:** The relationship between quality of life in women with osteoporosis and age, BMI, vitamin D levels, and pain.

QUALEFFO-41 Domains	Age	BMI	25(OH)D	Pain Score
			r	
Activities of daily living	0.41 *	−0.27 *	−0.12	0.48 *
Ability to do jobs around the house	0.59 *	−0.23 *	−0.11	0.51 *
Mobility	0.57 *	−0.28 *	−0.31 *	0.44 *
Pain	0.21 *	−0.31 *	−0.13	0.59 *
General health perception	0.18 *	−0.12 *	−0.13 *	0.45 *
Leisure and social activities	0.32 *	−0.24 *	−0.22 *	0.37 *
Mental function	0.49 *	−0.18 *	−0.29 *	0.51 *
Total	0.52 *	−0.21 *	−0.24 *	0.47 *

* *p* < 0.05.

## Data Availability

Publicly available datasets were not analyzed in this study.

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
