# Peer review of "Dietary Behaviors, Serum 25(OH)D Levels and Quality of Life in Women with Osteoporotic Disorders"

_ijerph, 2022, doi:10.3390/ijerph192417023_

Round 1

Reviewer 1 Report

Dear Editor,

thanks so much for the opportunity to revise the work entitled "Dietary Behaviors, Serum 25(OH)D levels and Quality of Life in Women with Osteoporotic Disorders" submitted by Dr Godala et al.

The work is very interesting, showing as quality of life could be strongly reduced in women affected by osteoporosis and related also to nutritional behaviors and vitamin D levels.

The paper is well written, the results clearly reported and the statistical methods rigorous.

I have not specific revisions for the authors to perform. I only suggest to mention, in introduction or discussion section, the negative effect of osteoporotic fractures on respiratory function and impairment, as recently reported, that could be surely and negatively influence the qol of patients affected ( DOI: 10.1007/s12020-022-03096-7 ) and as vitamin D was recently reported as a risk factor not only for osteoporosis but also for body mass alterations including sarcopenia and obesity in the novel described osteosarcopenic obesity phenotype ( DOI: 10.3390/nu14091816 ).

Thanks.

Author Response

Dear Reviewer,

Thank you for giving us the opportunity to submit a revised draft of the manuscript. We
appreciate the time and effort that you dedicated to providing feedback on our
manuscript and are grateful for the insightful comments on and valuable improvements to our paper. We have incorporated most of the suggestions made by the reviewers. Those changes are highlighted within the manuscript.

We have added information about the negative effect of osteoporotic fractures on respiratory function and impairment, as recently reported, that could be surely and negatively influence the qol of patients affected and as vitamin D was recently reported as a risk factor not only for osteoporosis but also for body mass alterations including sarcopenia and obesity in the novel described osteosarcopenic obesity phenotype. We have enriched the bibliography with suggested articles.

Best regards,

Authors

Reviewer 2 Report

Thank you for the paper entitled 'Dietary Behaviours, Serum 24(OH)D levels and Quality of Life in Women with Osteoporotic Disorders. The paper is interesting, however some key explanatory information is missing from the manuscript. Some key points of consideration are listed below:

Abstract

1. Abstract could more clearly demonstrate the gap in the literature and tell the story. In particular, I feel like the justification for investigate diet and vitamin D were poorly described.

2. Some details about the methodology could be relaxed in the abstract, especially where they are complicated to explain, such as the diet groups.

Introduction

3. Overall the introduction is strong, however some additional details to build the case for investigating diet could be provided. This would help guide the reader from the background to the aim of the study and explain the literature gap you are trying to fill.

Methods

4. More details need to be provided about the recruitment procedure. How were participants selected? How were controls selected? Response rates to selection process? Etc

5. The authors should also describe how participants were defined as having osteoporosis (ISCD criteria performed in clinic? Self report?)

6. The authors should note in the statistics section of the methods if adjustment for multiple comparisons was undertaken, as there are a number of analyses being presented in this study.

Results

7. The results section is detailed, however there is some redundancy in tables, for instance there does not need to be a line for both "yes" and "no" for previous fracture, as one implies the values for the other. 

8. Is it of value to report the specific frequency of consumption of each of the food groups in the dietary survey? Does this provide meaningful information to the reader?

Discussion

9. Be careful not to overstate the meaning of results. For instance, low quality of life was driven by the "mental function" domain, but the research presented in this paper does not provide sufficient data to infer "the emotional impact of the diagnosis and the patient's mental state". 

10. Overall, the discussion could be more concise. Particularly in the section exploring previous literature, where many of the reported results show similar associations.

11. I would also have liked to have seen more of a discussion about the interplay between Vitamin D status and osteoporosis and how this might have influenced results.

12. The section discussing the associations of pain and quality of life are interesting, particularly considering osteoporosis in theory is a symptomless condition. The possible reasons for this result could also be investigated in further depth. 

13. A general expansion of the strengths and limitations (especially in reflection to the recruitment procedure) would be of benefit.

Author Response

Dear Reviewer,

Thank you for giving us the opportunity to submit a revised draft of the manuscript. We
appreciate the time and effort that you dedicated to providing feedback on our
manuscript and are grateful for the insightful comments on and valuable improvements to our paper. We have incorporated most of the suggestions made by the reviewers. Those changes are highlighted within the manuscript.

Abstract

  1. Abstract could more clearly demonstrate the gap in the literature and tell the story. In particular, I feel like the justification for investigate diet and vitamin D were poorly described.

With all due respect, in our opinion the gap in the literature is clearly demonstrated.

  1. Some details about the methodology could be relaxed in the abstract, especially where they are complicated to explain, such as the diet groups.

We enriched the methodology section with mentioned details.

Introduction

  1. Overall the introduction is strong, however some additional details to build the case for investigating diet could be provided. This would help guide the reader from the background to the aim of the study and explain the literature gap you are trying to fill.

We made the introduction a little bit longer hoping that will satisfy the reviewers.

Methods

  1. More details need to be provided about the recruitment procedure. How were participants selected? How were controls selected? Response rates to selection process? Etc

We added the information about the recruitment procedure.

  1. The authors should also describe how participants were defined as having osteoporosis (ISCD criteria performed in clinic? Self report?)

We added the information about diagnostic criteria for osteoporosis.

  1. The authors should note in the statistics section of the methods if adjustment for multiple comparisons was undertaken, as there are a number of analyses being presented in this study.

No, it wasn’t.

Results

  1. The results section is detailed, however there is some redundancy in tables, for instance there does not need to be a line for both "yes" and "no" for previous fracture, as one implies the values for the other. 

We deleted the “no” lines.

  1. Is it of value to report the specific frequency of consumption of each of the food groups in the dietary survey? Does this provide meaningful information to the reader?

We deleted the part concerning food groups consumption.

Discussion

  1. Be careful not to overstate the meaning of results. For instance, low quality of life was driven by the "mental function" domain, but the research presented in this paper does not provide sufficient data to infer "the emotional impact of the diagnosis and the patient's mental state".

We deleted the confusing part of this statement.  

  1. Overall, the discussion could be more concise. Particularly in the section exploring previous literature, where many of the reported results show similar associations.

According to our best knowledge we discussed all relevant articles.

  1. I would also have liked to have seen more of a discussion about the interplay between Vitamin D status and osteoporosis and how this might have influenced results.

We have tried to make it more clear hoping it will be satisfying.

  1. The section discussing the associations of pain and quality of life are interesting, particularly considering osteoporosis in theory is a symptomless condition. The possible reasons for this result could also be investigated in further depth. 

We have tried to make it more clear hoping it will be satisfying.

  1. A general expansion of the strengths and limitations (especially in reflection to the recruitment procedure) would be of benefit.

With all due respect, in our opinion it is well balanced.

Best regards,

Authors
